# The Risk of Endoscopy-Related Bleeding in Patients with Liver Cirrhosis: A Retrospective Study

**DOI:** 10.3390/medicina59010170

**Published:** 2023-01-14

**Authors:** Su Bee Park, Jung Won Jeon, Hyun Phil Shin

**Affiliations:** Division of Gastroenterology, Department of Internal Medicine, Kyung Hee University Hospital at Gang Dong, Kyung Hee University College of Medicine, Seoul 05278, Republic of Korea

**Keywords:** liver cirrhosis, endoscopic procedure, bleeding, fibrosis

## Abstract

*Background and Objectives*: The risk of bleeding after endoscopic procedures in patients with liver cirrhosis remains unclear because of secondary blood coagulation disorders. In this study, we used various indices to evaluate the risk of bleeding in patients with cirrhosis. *Materials and Methods*: Patients with liver cirrhosis aged ≥18 years who underwent endoscopic interventions at Kyung Hee University Hospital at Gangdong between February 2007 and September 2021 were included. Clinical information, including demographic data, laboratory results, the presence of cirrhosis complications, and the degree of fibrosis, was checked and evaluated based on medical records. *Results*: A total of 101 patients with cirrhosis were analyzed. A total of 16 of the 101 patients (15.8%) experienced bleeding after the endoscopic procedure. One patient (0.99%) presented with spurting, while the others presented with mild oozing. All patients underwent hemostatic procedures using hemoclips. The presence of a varix significantly increased post-endoscopic bleeding (*p* = 0.03). Patients with FIB > 3.25 showed a statistically significant bleeding tendency (*p* = 0.00). *Conclusions*: There was no significant difference in bleeding risk according to the platelet count, prothrombin time, Child–Pugh score, and model for end-stage liver disease (MELD). Considering the degree of liver fibrosis and the invasiveness of the planned procedure, most endoscopic procedures can be performed safely but should be further evaluated in a cohort with a larger sample size.

## 1. Introduction

The risk of bleeding due to secondary blood coagulation disorders should be considered in patients with liver cirrhosis, and active endoscopic treatment can be challenging owing to bleeding concerns [1,2,3,4]. Although the life expectancy of patients after being diagnosed with cirrhosis is limited, it has been reported that the life expectancy of patients after being diagnosed with compensated cirrhosis is approximately 9–12 years [5,6], and even in patients with decompensated cirrhosis, the overall 5-year survival rate is approximately 45% [7]. Therefore, performing an endoscopic procedure at an appropriate time can further extend the life expectancy of patients with cirrhosis.

In cirrhosis, decreased levels of most procoagulant factors are accompanied by decreased antithrombin and anticoagulant levels [8]. Coagulation in patients with cirrhosis is balanced by these opposing drivers [9]. However, several factors should be considered when evaluating the risk of bleeding or thrombosis. Patients with liver cirrhosis may experience bleeding because of a non-coagulopathic mechanism [10]. The hemodynamic pathway of advanced cirrhosis is a significant cause of bleeding; however, laboratory tests for coagulation have been commonly used to assess this risk.

The risk of bleeding during endoscopic procedures in patients with cirrhosis has been studied previously [11,12,13]. It has been reported that colonoscopic polypectomy can be safely performed even in patients with cirrhosis, and the risk of major complications is relatively low [14]. However, another study reported that the Child–Pugh score was an independent risk factor for bleeding during colonoscopic polypectomy and that the risk of immediate bleeding after polypectomy was high for patients with cirrhosis in the Child B and Child C groups [15]. Several studies on the upper gastrointestinal tract in cirrhotic patients have reported various bleeding rates during endoscopic submucosal dissection procedures, ranging from 4.3% to 16.7% [16,17,18]. It is not easy to determine the safety issues associated with endoscopic procedures in patients with cirrhosis. Patients with cirrhosis are usually excluded from the general guidelines regarding transfusion before invasive procedures, and an appropriate evaluation of the risk is needed rather than prophylactic transfusion [19].

This study aimed to evaluate the risk of bleeding after various endoscopic procedures in patients with cirrhosis, using several more indices than previous studies. The model for end-stage liver disease (MELD), the aspartate aminotransferase (AST)-to-platelet ratio index (APRI), fibrosis-4 (FIB-4), and Child–Pugh scores were used to obtain more evidence.

## 2. Materials and Methods

Patients with liver cirrhosis aged ≥18 years who underwent endoscopic intervention at Kyung Hee University Hospital in Gangdong from February 2007 to September 2021 were selected, and complications and blood test results were reviewed. Patients with missing medical records for endoscopic procedures, those who were transferred to another hospital after endoscopic procedures, and those who could not be followed up were excluded from the study.

The following data were collected based on hospital medical records: age, sex, blood cell and platelet counts, the international normalized ratio (INR) of prothrombin time, and other serum biochemical test results. The complications of cirrhosis were evaluated as the clinical features of ascites, varices, and hepatic encephalopathy [20]. Hepatocellular carcinoma (HCC) was diagnosed if definite features of the hepatoma were confirmed by computed tomography or magnetic resonance imaging. Bleeding was defined as the continuous oozing of blood and active spurting immediately after the endoscopic procedure. Cases in which prophylactic endoscopic intervention was performed because the lesion was large or bleeding was expected were not counted as true hemostasis. The severity of fibrosis was evaluated according to the Child–Pugh score, MELD, APRI, and FIB-4 score [21,22,23]. The formula for the MELD is 3.78 × log_e_ [serum bilirubin (mg/dL)] + 11.2 × ln [INR] + 9.57 × log_e_ [serum creatinine (mg/dL)] + 6.43. The APRI was calculated as AST level (IU/L)/upper limit of normal AST (IU/L)/platelet count (10^9^/L) × 100, and the FIB-4 index was calculated as age (years) × AST (IU/L)/[platelet count (10^9^/L) × alanine aminotransferase^1/2^ (IU/L)]. The differences between the group with bleeding after endoscopic intervention and the group without bleeding were compared. For colonoscopic polypectomy, which had the highest number of endoscopic procedures, a comparison was made between the bleeding and non-bleeding groups, and any changes according to the size and shape of the polyp were evaluated. Gross polyp morphology was classified using the Paris endoscopic classification [24].

Continuous variables were expressed as mean ± standard deviation, and categorical variables were expressed as numbers and percentages. The differences between the bleeding and non-bleeding groups were analyzed using the *t*-test, and the Wilcoxon rank-sum test was used as a non-parametric method for the independent *t*-test. In this case, the median values and interquartile range results are also described. The relative risk was evaluated by dividing the MELD by ‘11,’ FIB-4 by ‘3.25’ [25], and the APRI by ‘1’ [26]. The optimal cutoff value for the MELD and polyp size was set as the point at which the sum of the sensitivity and specificity of the receiver operating characteristic curve was at its maximum. Multivariable logistic regression was defined as variables with a univariate *p*-value > 0.15. The software used for the analysis was SAS/STAT software (SAS 9.4; SAS Institute, Cary, NC, USA) and R 4.0.3 (R Foundation for Statistical Computing).

## 3. Results

### 3.1. Patients’ Clinical Characteristics 

A total of 101 patients were included in this study. Table 1 shows the clinical characteristics of patients with liver cirrhosis who underwent endoscopic intervention. The mean participant age was 58.48 ± 8.73 years, and 93 patients were male (92.1%). The endoscopic interventions included colonoscopic polypectomy, endoscopic mucosal resection, retrograde cholangiopancreatography, stomach endoscopic mucosal resection, and stomach endoscopic submucosal resection. The most common procedure was colonoscopic polypectomy (85 patients; 84.2%). Regarding the etiology of liver cirrhosis, alcohol was the most common cause, with 98 cases (97.0%), and cryptogenic cirrhosis and hepatitis B virus cirrhosis were the causative factors in 2 cases and 1 case, respectively. The mean serum total bilirubin of the patients was 1.87 ± 1.96 mg/dL, the mean albumin was 3.63 ± 0.66 g/dL, the mean prothrombin time (INR) was 1.25 ± 0.37, and the mean platelet count was 130.75 ± 60.48 × 10^3^/μL). Varices were the most common complication (45 cases, 44.6%), and there was no hepatic encephalopathy. Three patients had hepatocellular carcinoma (HCC). The number of Child A patients was the highest (68 patients (67.3%)), followed by Child B (26 patients (25.7%)), and Child C (seven patients (6.93%)) patients. The mean MELD was 10.64 ± 4.05, the mean FIB-4 score was 6.69 ± 6.18, and the mean APRI was 2.12 ± 2.63, respectively. A total of 16 out of 101 patients (15.8%) experienced bleeding after the endoscopic procedure. Among these 16 patients, 1 (0.99%) presented with spurting, while the others had oozing. All patients underwent hemostatic procedures using hemoclips.

### 3.2. Comparison between Bleeding and Non-Bleeding Groups

A comparison of the bleeding and non-bleeding groups (Table 2) revealed that there was no difference in age or sex and no difference in bleeding risk according to laboratory results, including platelet count and prothrombin time. The presence of a varix significantly increased the probability of post-endoscopic bleeding (*p* = 0.03). In terms of evaluating indices for cirrhosis, there was no significant relationship between the Child–Pugh score and bleeding, even in Child–Pugh scores B and C. Additionally, no significant relationship was found between the MELD and APRI. However, there was a significant relationship between the FIB-4 score and significant bleeding (*p* = 0.02). Patients with an FIB-4 score > 3.25 showed a statistically significant bleeding tendency (*p* = 0.002). A multivariate analysis was performed on platelets, varices, and HCC, but no meaningful values were derived. Multivariate analysis for FIB > 3.25 showed a *p*-value of 0.04.

### 3.3. Sub-Analysis in the Colonoscopic Polypectomy–EMR Group

In a further sub-analysis of the colonoscopic polypectomy group (Table 3), no laboratory test results, cirrhosis-related complications, or serum fibrosis markers had a significant value, except for FIB-4. There was no difference in the platelet counts and INR values between the bleeding and non-bleeding groups, and there was no effect on the complications of cirrhosis, the Child–Pugh score, or MELD. There was a significant relationship between the FIB-4 score and significant bleeding (*p* = 0.04). Patients with an FIB-4 score > 3.25 showed a statistically significant bleeding tendency (*p* = 0.008).

### 3.4. Risk of Bleeding Depending on the Characteristics of Polyps

In the colonoscopic polypectomy group, the bleeding tendency was not affected by polyp size. Pedunculated polyps showed statistically significant bleeding compared to sessile polyps (*p* = 0.03) (Table 4).

## 4. Discussion

The liver is an important hematologic organ that synthesizes or eliminates several factors related to hemostasis and thrombosis. Patients with liver cirrhosis are prone to coagulopathy due to increased portal pressure and dysfunction of coagulation factors [27,28,29]. This makes liver specialists and endoscopists cautious when performing invasive procedures in patients with cirrhosis. However, coagulopathy in liver disease is complex. Patients with cirrhosis are characterized by a clinical bleeding tendency and decreased levels of most procoagulant factors, except for elevated factor VIII and von Willebrand factor (vWF) [8]. However, a concomitant decrease in protein C, protein S, antithrombin, and fibrinolytic factors, which are natural anticoagulants, results in some balance at the same time [9]. 

Therefore, it is challenging to accurately predict the clotting status of patients with cirrhosis simply using the prothrombin time and platelet count [27,28,29]. Our study showed no difference in bleeding risk according to commonly used laboratory results, including platelet count and prothrombin time. However, in patients with advanced liver disease, hemorrhage is the main complication, accounting for 20% of deaths [30]; therefore, risk assessment using various tools is required to conduct invasive procedures in patients with liver cirrhosis [2,21,31]. Four markers, the Child–Pugh score, MELD, APRI, and FIB-4, were used to assess the risk of bleeding. The Child–Pugh score and MELD have been widely used to evaluate liver function and prognosis in patients with cirrhosis, and in many studies, both scores have shown almost similar evaluating power [32].

The APRI and FIB-4 are serum markers developed for diagnosing liver fibrosis and are calculated using liver function test values and platelet counts. In the case of FIB-4, an age factor is added, and it is known that FIB-4 exhibits similar or higher diagnostic accuracy than the APRI [25]. Both markers are useful for diagnosing advanced fibrosis and cirrhosis; however, unlike the Child–Pugh score and MELD, they cannot predict liver function and prognosis [25,26,33]. In our study, patients with FIB > 3.25 showed a statistically significant bleeding tendency; however, we are cautious in considering that FIB-4 is a more appropriate marker for predicting bleeding in cirrhotic patients than the three other markers.

As previously described, in patients with chronic liver disease, even if severe blood coagulation abnormalities are observed in the laboratory findings, actual bleeding symptoms are often not observed. This phenomenon occurs because the abnormality in laboratory performance, reflecting the deficiency of the blood clotting factor commonly observed in chronic liver disease, is balanced by an increase in factor VIII and vWF, which are currently not reflected in the simple laboratory coagulation test performed [34]. In addition, the quantitative and qualitative decrease in platelets is balanced by the decrease in disintegrins and metalloproteinase with a thrombospondin type 1 motif, member 13 (ADAMTS-13), and the increase in vWF, and the fibrinolytic component is also balanced because the hemostatic mechanism is rebalanced [34]. This can be seen in clinical practice, where hemostasis is achieved without plasma transfusion in most patients with cirrhosis, despite severe coagulation abnormalities during liver transplantation. This is supported by research showing that actual blood clotting is balanced even in cirrhosis of the liver [35]. Undoubtedly, patients with liver cirrhosis show a clear tendency to bleed; however, evaluating this tendency as a simple blood test result or as a single marker should be reconsidered. At the same time, it is unnecessary and sometimes even harmful to overcorrect the coagulation abnormality in the existing test results by being overly wary of the bleeding risk [36,37].

In our study, 85 of 101 patients (84.2%) underwent colonoscopic polypectomy. Bleeding risk according to the type of endoscopic intervention was not evaluated; however, in the sub-analysis of the colonoscopic polypectomy group, it is noteworthy that the bleeding risk was more affected by the pedunculated shape than the size.

Our study had several limitations. The number of patients participating in the study was as small as 101, and the etiology of cirrhosis was biased. It was impossible to accurately determine the number of patients who had lesions but did not undergo this procedure. Doctors usually perform invasive procedures carefully in Child B and C patients, and the number of these patients was not sufficient in our study. In a retrospective study, the FIB-4 score and APRI were used as convenient non-invasive fibrosis tests, but these were not enough to reflect the bleeding tendency. Moreover, the lack of testing for fibrinogen and other coagulation factors is a limitation. Because the number of included Child–Pugh B and C patients was small, the conclusion that therapeutic endoscopy is possible in all patients with liver cirrhosis requires caution. Further large-scale prospective multicenter studies are required to verify the risk of bleeding after various endoscopic procedures in patients with liver cirrhosis.

## 5. Conclusions

The bleeding risk associated with various endoscopic procedures in patients with liver cirrhosis is acceptably low and non-severe. There was no significant difference in bleeding risk according to platelet count, prothrombin time, Child–Pugh score, or the MELD. Among the markers, the statistically significant increase in bleeding when the FIB-4 score exceeded 3.25 is worth considering. Considering the degree of liver fibrosis and the invasiveness of the planned procedure, most endoscopic procedures can be performed safely; however, caution is needed in cases of decompensated liver cirrhosis. This conclusion should be further evaluated in a cohort study with a larger sample size.

## Figures and Tables

**Table 1 medicina-59-00170-t001:** Clinical characteristics of patients with liver cirrhosis who underwent the endoscopic procedure.

Variable	All Patients (N = 101)
Age (years)	58.48 ± 8.73
Gender	
Male	93 (92.08%)
Female	8 (7.92%)
Type of endoscopic intervention	
Colonoscopic polypectomy–endoscopic mucosal resection	85 (84.16%)
Endoscopic retrograde cholangiopancreatography	9 (8.91%)
Stomach endoscopic mucosal resection	2 (1.98%)
Stomach endoscopic submucosal resection	5 (4.95%)
Etiology of LC	
Alcohol	98 (97.03%)
HBV	1 (0.99%)
Cryptogenic	2 (1.98%)
Histology	
Benign	98 (97.03%)
Malignant	3 (2.97%)
Serum total bilirubin (mg/dL)	1.87 ± 1.96
Serum AST (IU/L)	76.45 ± 63.87
Serum ALT (IU/L)	46.30 ± 43.25
Serum albumin (g/dL)	3.63 ± 0.66
Serum protein (g/dL)	7.07 ± 0.85
Platelet count (×10^3^/㎕)	130.75 ± 60.48
Prothrombin time (INR)	1.25 ± 0.37
Hemoglobin (g/dL)	12.11 ± 2.54
Serum AFP (ng/mL)	11.66 ± 48.03
Serum cholesterol (mg/dL)	147.62 ± 56.27
Serum creatinine (mg/dL)	0.95 ± 0.36
Serum sodium (mEq/L)	134.46 ± 19.61
Complications of cirrhosis	
Varices (+)	45 (44.55%)
Ascites (+)	26 (26.73%)
Hepatic encephalopathy (+)	0 (0.99%)
HCC present	3 (2.97%)
Child–Pugh class	
A	68 (67.33%)
B	26 (25.74%)
C	7 (6.93%)
MELD	10.64 ± 4.05
FIB-4	6.69 ± 6.18
APRI	2.12 ± 2.63
Bleeding after endoscopic procedure	
Spurting	1 (0.99%)
Oozing	15 (14.85%)

Values are expressed as mean ± SD or frequency (%). Abbreviations: LC, liver cirrhosis; HBV, hepatitis B virus; HCC, hepatocellular carcinoma; MELD model for end-stage liver disease; AST aspartate aminotransferase; ALT alanine aminotransferase; INR international normalized ratio; AFP, alpha-fetoprotein.

**Table 2 medicina-59-00170-t002:** Clinical characteristics of patients with LC who underwent endoscopic intervention.

	Patients Who Did Not Develop Post-Endoscopic Bleeding (N = 85)			Patients Who Did Develop Post-Endoscopic Bleeding (N = 16)				Univariable Firth’s Logistic Regression			
		Median	IQR		Median	IQR	*p*-Value	Odds Ratio	95% CI		*p*-Value
Age (years)	58.71 ± 8.96	59.00	13.00	57.25 ± 7.51	57.50	9.00	0.52	0.92	0.92	1.04	0.55
Gender											
Male	79 (92.94%)			14 (87.50%)			0.60 *	0.44	0.09	2.41	0.38
Female	6 (7.06%)			2 (12.50%)				.	.	.	.
Etiology of LC							>0.99 *				
Alcohol	82 (96.47%)			16 (100.00%)				.	.	.	.
HBV	1 (1.18%)			0 (0.00%)				1.73	0.01	158.50	0.83
Cryptogenic	2 (2.35%)			0 (0.00%)				1.00	0.02	42.80	0.90
Serum total bilirubin (mg/dL)	1.91 ± 2.07	1.20	1.20	1.66 ± 1.26	1.35	1.10	0.90	0.93	0.73	1.29	0.85
Serum AST (IU/L)	76.36 ± 66.56	58.00	59.00	76.88 ± 48.88	54.00	59.50	0.44	1.01	0.99	1.01	0.82
Serum ALT (IU/L)	48.92 ± 45.80	36.00	29.00	32.38 ± 21.88	28.00	15.50	0.12	0.97	0.97	1.01	0.23
Serum albumin (g/dL)	3.65 ± 0.65	3.70	0.95	3.54 ± 0.70	3.70	0.90	0.57	0.76	0.35	1.69	0.51
Serum protein (g/dL)	7.04 ± 0.85	7.00	1.10	7.23 ± 0.87	7.35	1.25	0.34	1.32	0.69	2.47	0.42
Platelet count (×10^3^/㎕)	135.02 ± 62.38	125.00	86.00	108.06 ± 44.04	98.50	70.50	0.11	0.92	0.98	1.00	0.12
Prothrombin time (INR)	1.26 ± 0.40	1.13	0.31	1.20 ± 0.14	1.21	0.16	0.55	0.85	0.18	3.82	0.81
Hemoglobin (g/dL)	12.09 ± 2.58	12.40	3.60	12.19± 2.39	12.15	3.05	0.94	1.03	0.82	1.25	0.90
Serum AFP (ng/mL)	12.71 ± 52.64	5.30	5.78	6.48 ± 3.50	5.11	5.35	0.59	1.01	0.99	1.01	0.85
Serum cholesterol (mg/dL)	148.03 ± 60.33	145.00	59.00	145.44 ± 26.79	149.50	34.50	0.69	1.00	0.99	1.01	0.97
Serum creatinine (mg/dL)	0.95 ± 0.37	0.90	0.35	0.90 ± 0.28	0.81	0.26	0.59	0.78	0.15	3.71	0.72
Serum sodium (mEq/L)	135.36 ± 15.70	137.00	5.00	129.84 ± 33.60	138.00	6.50	0.79	0.99	0.97	1.01	0.31
Complications of cirrhosis											
Varices (+)	34 (40.00%)			11 (68.75%)			0.03	3.11	1.02	9.51	0.045
Ascites (+)	23 (27.06%)			4 (25.00%)			>0.99 *	0.97	0.29	3.16	0.94
HCC present	1 (1.18%)			2 (12.50%)			0.065 *	9.74	0.88	107.47	0.064
Child–Pugh class							0.73 *				
A	56 (65.88%)			12 (75.00%)				.	.	.	.
B	22 (25.88%)			4 (25.00%)				0.94	0.27	3.01	0.60
C	7 (8.24%)			0 (0.00%)				0.31	0.01	6.84	0.47
MELD	10.75 ± 4.25	10.00	6.00	10.06 ± 2.84	10.00	3.00	0.86	0.95	0.84	1.11	0.61
MELD > 11	46	54.12		11	68.75		0.28	.	.	.	0.31
MELD ≤ 11	39	45.88		5	31.25			0.53	0.19	1.71	
FIB-4	6.47 ± 6.51	4.53	5.02	7.85 ± 3.95	7.66	5.56	0.02	1.06	0.96	1.12	0.35
FIB-4 > 3.25	33 (38.82%)			0 (0.00%)			0.002	.	.	.	0.04
FIB-4 ≤ 3.25	52 (61.18%)			16 (100.00%)				21.06	1.17	378.51	
APRI	2.12 ± 2.80	1.26	1.57	2.07 ± 1.50	1.52	1.61	0.21	1.02	0.84	1.23	0.85
APRI > 1	37 (43.53%)			4 (25.00%)			0.17	.	.	.	0.20
APRI ≤ 1	48 (56.47%)			12 (75.00%)				2.15	0.67	6.93	

Values are expressed as mean ± SD or frequency (%), chi-square test or Fisher’s exact (*).

**Table 3 medicina-59-00170-t003:** Clinical characteristics of patients with LC who underwent colonoscopic polypectomy–EMR (N = 85).

	Patients Who Did Not Develop Post-Endoscopic Bleeding (N = 85)			Patients Who Did Develop Post-Endoscopic Bleeding (N = 16)				Univariable Firth’s Logistic Regression			
		Median	IQR		Median	IQR	*p*-Value	Odds Ratio	95% CI		*p*-Value
Age (years)	58.71 ± 8.96	59.00	13.00	57.25 ± 7.51	57.50	9.00	0.52	0.92	0.92	1.04	0.55
Gender											
Male	79 (92.94%)			14 (87.50%)			0.60 *	0.44	0.09	2.41	0.38
Female	6 (7.06%)			2 (12.50%)				.	.	.	.
Etiology of LC							>0.99 *				
Alcohol	82 (96.47%)			16 (100.00%)				.	.	.	.
HBV	1 (1.18%)			0 (0.00%)				1.73	0.01	158.50	0.83
Cryptogenic	2 (2.35%)			0 (0.00%)				1.00	0.02	42.80	0.90
Serum total bilirubin (mg/dL)	1.91 ± 2.07	1.20	1.20	1.66 ± 1.26	1.35	1.10	0.90	0.93	0.73	1.29	0.85
Serum AST (IU/L)	76.36 ± 66.56	58.00	59.00	76.88 ± 48.88	54.00	59.50	0.44	1.01	0.99	1.01	0.82
Serum ALT (IU/L)	48.92 ± 45.80	36.00	29.00	32.38 ± 21.88	28.00	15.50	0.12	0.97	0.97	1.01	0.23
Serum albumin (g/dL)	3.65 ± 0.65	3.70	0.95	3.54 ± 0.70	3.70	0.90	0.57	0.76	0.35	1.69	0.51
Serum protein (g/dL)	7.04 ± 0.85	7.00	1.10	7.23 ± 0.87	7.35	1.25	0.34	1.32	0.69	2.47	0.42
Platelet count (×10^3^/㎕)	135.02 ± 62.38	125.00	86.00	108.06 ± 44.04	98.50	70.50	0.11	0.92	0.98	1.00	0.12
Prothrombin time (INR)	1.26 ± 0.40	1.13	0.31	1.20 ± 0.14	1.21	0.16	0.55	0.85	0.18	3.82	0.81
Hemoglobin (g/dL)	12.09 ± 2.58	12.40	3.60	12.19 ± 2.39	12.15	3.05	0.94	1.03	0.82	1.25	0.90
Serum AFP (ng/dL)	12.71 ± 52.64	5.30	5.78	6.48 ± 3.50	5.11	5.35	0.59	1.01	0.99	1.01	0.85
Serum cholesterol (mg/dL)	148.03 ± 60.33	145.00	59.00	145.44 ± 26.79	149.50	34.50	0.69	1.00	0.99	1.01	0.97
Serum creatinine (mg/dL)	0.95 ± 0.37	0.90	0.35	0.90 ± 0.28	0.81	0.26	0.59	0.78	0.15	3.71	0.72
Serum sodium (mEq/L)	135.36 ± 15.70	137.00	5.00	129.84 ± 33.60	138.00	6.50	0.79	0.99	0.97	1.01	0.31
Complications of cirrhosis											
Varices (+)	34 (40.00%)			11 (68.75%)			0.03	3.11	1.02	9.51	0.045
Ascites (+)	23 (27.06%)			4 (25.00%)			>0.99 *	0.97	0.29	3.16	0.94
HCC present	1 (1.18%)			2 (12.50%)			0.065 *	9.74	0.88	107.47	0.064
Child–Pugh class							0.73 *				
A	56 (65.88%)			12 (75.00%)				.	.	.	.
B	22 (25.88%)			4 (25.00%)				0.94	0.27	3.01	0.60
C	7 (8.24%)			0 (0.00%)				0.31	0.01	6.84	0.47
MELD	10.75± 4.25	10.00	6.00	10.06 ± 2.84	10.00	3.00	0.86	0.95	0.84	1.11	0.61
MELD > 11	46	54.12		11	68.75		0.28	.	.	.	0.31
MELD ≤ 11	39	45.88		5	31.25			0.53	0.19	1.71	
FIB-4	6.47 ± 6.51	4.53	5.02	7.85 ± 3.95	7.66	5.56	0.02	1.06	0.96	1.12	0.35
FIB-4 > 3.25	33 (38.82%)			0 (0.00%)			0.002	.	.	.	0.04
FIB-4≤ 3.25	52 (61.18%)			16 (100.00%)				21.06	1.17	378.51	
APRI	2.12 ± 2.80	1.26	1.57	2.07 ± 1.50	1.52	1.61	0.21	1.02	0.84	1.23	0.85
APRI > 1	37 (43.53%)			4 (25.00%)			0.17	.	.	.	0.20
APRI≤ 1	48 (56.47%)			12 (75.00%)				2.15	0.67	6.93	

Values are expressed as mean ± SD or frequency (%). * Fisher’s exact was used.

**Table 4 medicina-59-00170-t004:** Polyp-related characteristics in post-colonoscopic polypectomy bleeding (N = 85).

	Post-Endoscopic Non-Bleeding (N = 73)			Post-Endoscopic Bleeding (N = 13)				Univariable Firth’s Logistic Regression			
		Median	IQR		Median	IQR	*p*-Value	Odds Ratio	95% CI		*p*-Value
Size	9.44 ± 5.33	8.00	3.00	12.00 ± 6.24	10.00	8.00	0.15	1.07	0.98	1.17	0.16
Size > 12 mm	60 (84.51%)			8 (61.54%)			0.12 *	.	.	.	0.06
Size ≤ 12 mm	11 (15.49%)			5 (38.46%)				3.4	0.95	12.16	
Gross morphology							0.03				
Sessile (Is)	46 (63.89%)			4 (30.77%)				.	.	.	0.04
Pedunculated (Ip)	26 (36.11%)			9 (69.23%)				3.71	1.08	12.71	

Values are expressed as mean ± SD or frequency (%). The polyp size was calculated as representative only of the largest polyp; chi-square test or Fisher’s exact (*) was used for categorical variables; Wilcoxon rank-sum test was used for continuous variables; univariable Firth logistic regression was performed with the non-bleeding group as a reference.

## Data Availability

The data published in this study are available upon request from the first author.

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
