# Peer review of "The Risk of Endoscopy-Related Bleeding in Patients with Liver Cirrhosis: A Retrospective Study"

_medicina, 2023, doi:10.3390/medicina59010170_

Round 1
Reviewer 1 Report
This study aiming at evaluated risk of bleeding after endoscopic procedures in cirrhotic patients is interesting , the reviewer has some questions due to the methodology the number of 101 patients other a 14 years period is low , a flow chart would be usefull to note how many patients were not studied and why , it is amazing that so many patients had viral hepatitis please comment ; moreover results should be interpreted cautiously due to the poor number of patients with B or C Child pugh cirrhosis
Author Response
Point 1: This study aiming at evaluated risk of bleeding after endoscopic procedures in cirrhotic patients is interesting , the reviewer has some questions due to the methodology the number of 101 patients other a 14 years period is low, a flow chart would be usefull to note how many patients were not studied and why, it is amazing that so many patients had viral hepatitis please comment ; moreover results should be interpreted cautiously due to the poor number of patients with B or C Child pugh cirrhosis
Response 1: Thank you for your valuable comments. I agree about the small number of patients in our study. There was no artificial exclusion criteria of patients. The reason for the small number of patients was that only cirrhotic patients who underwent invasive procedures were included. The number of Child B and C patients was small, because invasive procedures in patients with decompensated liver cirrhosis are determined by considering their life expectancy.
It is true that the etiology of liver cirrhosis is biased, which seems to be due to the difference in the frequency of endoscopic examinations depending on the etiology of liver cirrhosis.
Thanks to your valuble comments, I described more limitation you pointed out. The conclusion that therapeutic endoscopy is possible in all patients with liver cirrhosis definitely requires caution. I also clarified it in the conclusion section.
Reviewer 2 Report
Park et al. reported the bleeding risk during endoscopic procedure in patients with cirrhosis. It is needed to address several issues.
1. The important risk factors for coagulopathy and bleeding tendency during procedure are platelet count (<50,000/dL) and fibrinogen (<120 mg/dL)( (PMID: 25023035). However, this study did not check fibrinogen level. Add this limitation in the discussion section.
2. In the same manner, I think platelet count is the significant factor for bleeding during procedure. I recommend that platelet count would be dealt with categorical variable (e.g. platelet count >50,000 vs ≤50,000) rather than continuous variable. This study showed that FIB-4 index>3.25 is the risk factor for bleeding. Actually, FIB-4 consists of AST, ALT and platelet count. However, AST and ALT seem not to be associated with bleeding risk in cirrhosis among the components of FIB-4 index.
3. How many patients experience severe bleeding such as hypotension, shock, or blood transfusion?
Author Response
Point 1: The important risk factors for coagulopathy and bleeding tendency during procedure are platelet count (<50,000/dL) and fibrinogen (<120 mg/dL)( (PMID: 25023035). However, this study did not check fibrinogen level. Add this limitation in the discussion section.
Response 1: Thanks for your valuable point, I included this in the limitation section.
Point 2 : In the same manner, I think platelet count is the significant factor for bleeding during procedure. I recommend that platelet count would be dealt with categorical variable (e.g. platelet count >50,000 vs ≤50,000) rather than continuous variable. This study showed that FIB-4 index>3.25 is the risk factor for bleeding. Actually, FIB-4 consists of AST, ALT and platelet count. However, AST and ALT seem not to be associated with bleeding risk in cirrhosis among the components of FIB-4 index.
Response 2: Evaluating the risk of bleeding by dividing the groups based on platelet count 50000 could be reasonable. That was also in our options when the study was designed. However, a few patients appeared to have less than 50,000 counts. After discussing the statistics part, it was not included in the analysis considering its low statistical significance. In the limitation part, we specified the fact that a small number of Child B and C patients were included in the study.
In the case of FIB-4, I agree with your point and FIB-4 is less predictive for bleeding risk. However, as a non-invasive test for fibrosis, it can be easily used even when performing procedures in parts other than the hepatology. That’s the reason why we suggested. We added these contents also in the limitation.
Point 3 : How many patients experience severe bleeding such as hypotension, shock, or blood transfusion?
Response 3: There was only one case (0.99%) presented with spurting and well controlled by hemoclips without blood transfusion and none of the patients had experienced severe bleeding or shock.
Round 2
Reviewer 2 Report
The authors fully addressed the previously mentioned issues.